# CONDITIONAL LoRA PARAMETER GENERATION

## ABSTRACT

Generative models have achieved remarkable success in image, video, and text domains. Inspired by this, researchers have explored utilizing generative models to generate neural network parameters. However, these efforts have been limited by the parameter size and the practicality of generating high-performance parameters. In this paper, we propose COND P-DIFF, a novel approach that demonstrates the feasibility of controllable high-performance parameter generation, particularly for LoRA (Low-Rank Adaptation) weights, during the fine-tuning process. Specifically, we employ an autoencoder to extract efficient latent representations for parameters. We then train a conditional latent diffusion model to synthesize high-performing model parameters from random noise based on specific task conditions. Experimental results in both computer vision and natural language processing domains consistently demonstrate that COND P-DIFF can generate high-performance parameters conditioned on the given task. Moreover, we observe that the parameter distribution generated by COND P-DIFF exhibits differences compared to the distribution obtained through normal optimization methods, indicating a certain level of generalization capability. Our work paves the way for further exploration of condition-driven parameter generation, offering a promising direction for task-specific adaptation of neural networks.

## 1 INTRODUCTION

Recent advancements in generative models Rombach et al. (2022); Ramesh et al. (2022); Saharia et al. (2022); Brown et al. (2020) have marked substantial progress across several domains of artificial intelligence. In the computer vision domain, generative adversarial networks Goodfellow et al. (2014), diffusion models Ho et al. (2020), and other approaches Dinh et al. (2014); Rezende et al. (2014) have shown impressive results in image synthesis and manipulation. Notably, models such as Stable Diffusion Rombach et al. (2022), DALL-E 2 Ramesh et al. (2022), and Imagen Saharia et al. (2022) have set new benchmarks in the quality and resolution of generated images. Moreover, video generation models like Sora OpenAI (2024) have shown promising results in producing coherent and high-quality video sequences, opening new avenues for applications in entertainment and media. In the natural language processing domain Radford et al. (2019); Kaplan et al. (2020); Wei et al. (2022), autoregressive models like GPT Brown et al. (2020) and Llama Touvron et al. (2023) have demonstrated promising generation capabilities and alignment with human preference Jin et al. (2024); Ouyang et al. (2022); Rafailov et al. (2024); Kadavath et al. (2022), which underscore the potential of generative models.

Inspired by these achievements, recent studies Peebles et al. (2022); Wang et al. (2024) have begun to explore the application of generative models in novel areas, *generating high-performing model parameters*. These studies focus on directly generating novel model parameters to accelerate the training process, uncovering parameters that achieve comparable performance with those obtained through conventional optimization methods.

By harnessing the power of generative models, it is possible to substantially reduce the computational cost and time required for model optimization Peebles et al. (2022); Ruder (2016); Kingma & Ba (2014). Besides, examining the latent relationships between model parameters and performance provides valuable insights into the behavior and characteristics of neural networks Ha et al. (2016).

However, previous works on parameter generation Wang et al. (2024); Peebles et al. (2022); Soro et al. (2024); Schürholt et al. (2022); Knyazev et al. (2021) face several limitations. On the one hand,

the scale of parameters generated by prior methods Soro et al. (2024); Peebles et al. (2022); Wang et al. (2024) is insufficient for practical applications. For example, G.pt Peebles et al. (2022) has been evaluated only on relatively simple datasets such as MNIST and CIFAR-10, which may not sufficiently demonstrate its generalization ability when applied to more complex tasks, and p-diff (Wang et al., 2024) can generate small-scale high-performance parameters for simple architectures. Besides, Schürholt et al. (2022) learn a hyper-representation on model zoos for generative use to sample new small-scale model weights. On the other hand, previous methods do not support conditional high-performance parameter generation. P-diffWang et al. (2024) lacks support for conditional parameter generation, a crucial feature for real-world applications. Although G.pt Peebles et al. (2022) enables controllable parameter generation as an optimizer, it can hardly exhibit comparable performance compared to networks trained by conventional optimization methods.

Therefore, despite their promising potential, these methods grapple with constraints about parameter size, practicality, and overall performance, which yield the primary question to be explored in this paper: *(Q) Can we synthesize high-performance parameters conditioned on the given task practically?*

To enhance the practicality of parameter generation, two main challenges exist. First, parameter generation for complex models entails significant data preparation costs. For example, G.pt Peebles et al. (2022) requires training 23 million models, which is infeasible for large models. Second, controllable parameter generation is challenging due to the difficulty in modeling the distribution of parameters, making full parameter generation highly complex. Consequently, we focus on the conditional generation of fine-tuned LoRA (Low-Rank Adaptation) parameters in various domains as LoRA improves downstream task performance with few parameters and a relatively more stable distribution.

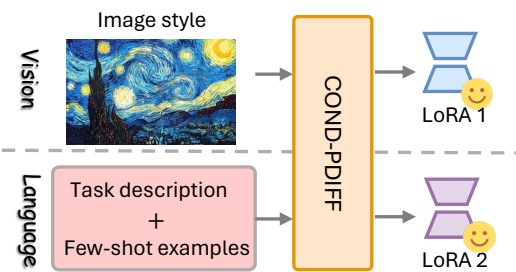

Figure 1: **High-performance LoRA parameters generation process by** COND P-DIFF **in vision and language domains.**

Specifically, LoRA Hu et al. (2021) is a parameter-efficient fine-tuning technique that adapts pre-trained models to specific tasks by learning low-rank matrices that modify the model's weights.

To achieve high-performance controllable conditional parameter generation, we propose Conditional Parameter Diffusion, named COND P-DIFF, which utilizes a standard latent diffusion model to perform conditional generation, synthesizing a new set of parameters tailored to specific conditions. Specifically, we use an autoencoder and a conditional latent diffusion model to capture the distribution of network weights. First, the autoencoder is trained on a selected set of parameters from models optimized with normal optimization methods, *e.g.*, SGD Ruder (2016), on different datasets, creating latent representations of these parameters. Second, we utilize a domain-specific condition, *e.g., text, style image*, projector to encode the condition information and train a conditional diffusion model to reconstruct latent representations. Finally, as shown in Figure 1, the trained conditional latent diffusion model COND P-DIFF generates latent representations from random noise in the inference process based on specific task conditions. Then, the decoder of the trained autoencoder processes these generated representations to produce new, high-performing model parameters.

Our method has the following characteristics: i) It demonstrates comparable or superior performance relative to models trained with conventional methods, spanning various datasets and architectures. ii) The parameters generated by our approach significantly differ from the parameters obtained during normal training, highlighting its capability to synthesize novel parameters rather than merely replicating the training examples. iii) Extensive evaluations demonstarte the robustness of our approach. Our method COND P-DIFF also shows generalizability in generated high-performance model weights space. We hope that our findings will provide new insights into the potential of applying conditional diffusion models to parameter generation and highlight a promising direction for task-specific parameter generation of neural networks.

## 2 PRELIMINARY

### 2.1 PRELIMINARIES OF LORA

Low-Rank Adaptation (LoRA) Hu et al. (2021) enhances the efficiency of fine-tuning large pre-trained language models by minimizing the computational demands usually required for full model retraining. LoRA introduces two trainable matrices, $B \in \mathbb{R}^{d \times r}$ and $A \in \mathbb{R}^{r \times k}$, to each transformer layer. These matrices, where $r$ is much smaller than hidden layer dimension $d$ and task-specific dimension $k$, perform a low-rank approximation of the typical updates made during fine-tuning. The core idea is that the necessary adjustments for task-specific adaptation have a low "intrinsic dimension," allowing significant reductions in trainable parameters while maintaining performance. The pretrained weight matrix $W_0$ remains unchanged, with only $B$ and $A$ being optimized, thus speeding up training and decreasing memory and computational needs. The modified forward pass in LoRA is represented as:

$$W_0 x + \Delta W x = W_0 x + B(Ax) \tag{1}$$

where $\Delta W = BA$ is the update. Initially, $B$ is zero, ensuring no changes to $W_0$, and $A$ starts with a small random Gaussian distribution. In deployment, the learned low-rank matrices $B$ and $A$ can be integrated into $W_0$. In this work, we aim to synthesize LoRA parameters because of the practicality and effective LoRA fusion that show the continuous distribution in LoRA parameter space.

### 2.2 PRELIMINARIES OF CONDITIONAL DIFFUSION MODELS

Conditional diffusion models Ho et al. (2020); Rombach et al. (2022); Zhang et al. (2023) extend the standard diffusion model by incorporating conditions into both the forward and reverse processes. This conditional information defined by $c$ allows the model to generate data tailored to specific attributes or requirements.

**Conditional forward process:** The forward process in conditional models involves adding noise to an initial sample while conditioning on $c$. The probability of transitioning from $x_{t-1}$ to $x_t$ under condition $c$ is modeled as a Gaussian distribution:

$$q(x_t|x_{t-1}, c) = \mathcal{N}(x_t; \sqrt{1 - \beta_t} x_{t-1}, \beta_t \mathbf{I}) \tag{2}$$

where $\beta_t$ are the timestep-dependent noise levels, and $\mathbf{I}$ represents the identity matrix. The complete forward process conditioned on $c$ is given by:

$$q(x_{1:T}|x_0, c) = \prod_{t=1}^{T} q(x_t|x_{t-1}, c) \tag{3}$$

**Conditional Reverse Process:** The reverse process aims to reconstruct the original sample from its noisiest state $x_T$ conditioned on $c$. It is formulated by:

$$p_\theta(x_{t-1}|x_t, c) = \mathcal{N}(x_{t-1}; \mu_\theta(x_t, t, c), \Sigma_\theta(x_t, t, c)) \tag{4}$$

In this process, $\mu_\theta$ and $\Sigma_\theta$ are functions estimated by a neural network, which also processes the condition $c$, ensuring that the recovery of data respects the conditional constraints.

**Optimization and Inference with Conditions:** The training procedure involves minimizing the Kullback-Leibler(KL) divergence between the forward and reverse conditional distributions, specifically:

$$L_{dm} = \mathbb{E}_{q(x_0, c)} \left[ D_{KL}(q(x_{t-1}|x_t, x_0, c) \| p_\theta(x_{t-1}|x_t, c)) \right] \tag{5}$$

During inference, the model generates new samples by conditioning on $c$ and sequentially applying the learned reverse transitions from a noise distribution, enabling the generation of data that closely adheres to the specified conditions.

## 3 METHODOLOGY

### 3.1 OVERVIEW

We propose conditional parameter generation to synthesize new parameters tailored to specific task conditions. Fig 2 illustrates our proposed COND P-DIFF framework. First, given a training dataset of

model parameters, we use an autoencoder Kingma & Welling (2013) to extract latent representations of the parameters and reconstruct the latent vectors by decoder. Then, inspired by Wang et al. (2024), we train a conditional latent diffusion model to generate high-performance parameters conditioned on specific task information. Finally, after training, we employ COND P-DIFF by feeding random noise and task-specific conditions into a conditional parameter diffusion model to generate the desired parameters.

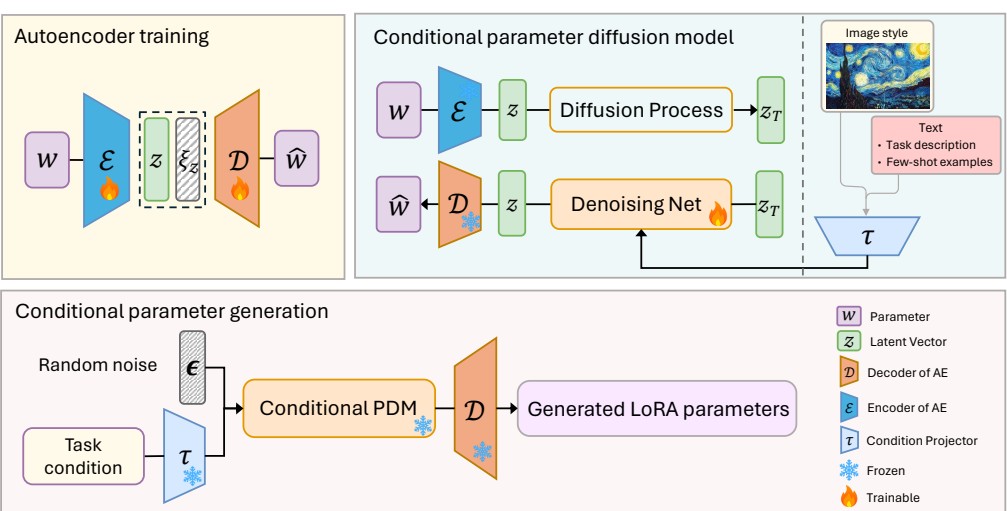

**Figure 2: The framework of COND P-DIFF. The autoencoder is employed to extract the latent representation of LoRA parameters and reduce memory consumption. The conditional parameter diffusion model aims to synthesize high-performance parameters based on specific task conditions.**

### 3.2 PARAMETER AUTOENCODER

**Dataset preparation.** In this work, we focus on synthesizing LoRA learnable matrix parameters of fine-tuned models by default. To obtain the training dataset for the parameter autoencoder, we fine-tune the pre-trained model using LoRA on the dataset for task $q$ and collect $N$ different checkpoints in the last $N$ steps. We denote the training dataset as $\Theta = [\theta_1, \ldots, \theta_n, \ldots, \theta_N]$, where $\theta_k$ represents the weights of LoRA for the model at a specific fine-tuning stage. Because the training dataset for COND P-DIFF contains model parameters rather than conventional image or language datasets, we propose *task normalization*. Specifically, we employ Z-Score normalization on the parameters of each task individually Ioffe & Szegedy (2015).

**Training procedure.** Given a training sample $\theta_n$, we flatten parameter matrix $\theta_n$ to a one-dimensional vector $w_n \in \mathbb{R}^{K \times 1}$, which $K$ is the total number of parameter weights of $w_n$. Then, we utilize an auto-encoder to obtain meaningful and robust latent representations. Specifically, we formulate the process as Equation 6, where $\mathcal{E}$ and $\mathfrak{D}$ represent the encoder and decoder functions, respectively. $z_n$ is the latent representation of the parameter matrix. $\hat{w}_n$ is the reconstruction of parameter $w_n$. To enhance the generalization and robustness of the autoencoder, we introduce Gaussian noise $\xi_\mathbf{z}$ to the latent vector. The final auto-encoder process is formulated as follows:

$$z_n = \mathcal{E}(w_n) = \text{Encoder}(w_n) \tag{6a}$$
$$\hat{w}_n = \mathfrak{D}(z_n) = \text{Decoder}(z_n + \xi_\mathbf{z}) \tag{6b}$$

We train the autoencoder function by minimizing loss function below.

$$\mathcal{L} = \frac{1}{N} \sum_{n=1}^{N} \|w_n - \hat{w}_n\|^2 \tag{7}$$

### 3.3 CONDITIONAL PARAMETER GENERATION

We utilize a conditional latent diffusion model to synthesize high-performance parameters based on conditions $y$ such as text and image. To handle different tasks and modalities, we adopt the domain-specific encoder, which is denoted as $\tau_{\text{domain}}(y; \rho)$, where $y$ represents the input condition and $\rho$ denotes the encoder parameters. For example, in the NLP experiments of this work, we employ the text decoder in CLIPRadford et al. (2021). Inspired by in-context learning, the input condition $y$ consists of a task description and two-shot examples to capture the task information. Besides, we utilize stylized images as conditions in style transfer tasks and adopt ResNet He et al. (2016) to extract style latent representations as the condition vector. More details about the condition are shown in Appendix 6.1. Regarding the U-Net architecture, we apply one-dimensional convolutions in denoising autoencoders because the weight matrix parameters do not show strong positional relationships different from images where pixels have two-dimensional spatial relationships.

Therefore, given the condition and training parameters samples, we train the conditional latent diffusion model through

$$L_{LDM} := \mathbb{E}_{\epsilon \sim \mathcal{N}(0,1),t} \left[ \|\epsilon - \epsilon_\theta(p_t, t, \tau_{domain,\rho}(y))\|^2 \right], \tag{8}$$

where $\epsilon_\theta$ is learned via Eq. 8. Finally, after conditional diffusion model training, we feed specific conditions corresponding to tasks and random noise to reverse the inference process to obtain high-performing weights for specific tasks.

## 4 EXPERIMENT

In this section, we first show the experiment setup. Then, we present the evaluation results, ablation studies, and analysis of COND P-DIFF.

### 4.1 EXPERIMENT SETUP

**Datasets and metrics.** We evaluate our method across various domains. Specifically, in NLP experiments, we test on the language understanding GLUE benchmark Wang et al. (2018). In CV experiments, we focus on the style-transfer tasks. We use the SemArt and WikiArt datasets Garcia & Vogiatzis (2018); Saleh & Elgammal (2015), which contain diverse artistic images, and evaluate them using the Fréchet Inception Distance (FID, Heusel et al. (2017), as employed by StyleGAN Karras et al. (2019), with lower scores indicating better performance.

**Dataset collecting and training procedures.** In NLP experiments, we collect 150 training samples for models, including BERT, Roberta, GPT-2 by default. For instance, in the case of BERT, we fixed pre-trained parameters and fine-tuned the network using LoRA. Specifically, we conduct the hyperparameter search for fixed values of $r$ and $\alpha$ and select the fine-tuning hyperparameters that yield the best average performance. During the fine-tuning process, we save the checkpoints of the last 150 steps as the training dataset, which includes the LoRA learnable matrix weights. In the framework of COND P-DIFF, the autoencoder includes 1D CNN-based encoders and decoders. We utilize the text encoder from CLIP as the condition text encoder. In image style transfer tasks, we fine-tune attention modules of a popular text-to-image model, PIXART-$\alpha$ model Chen et al. (2024) using LoRA and collected the last 64 LoRA checkpoints of the training process once in 10 steps. In the framework of COND P-DIFF, we used pre-trained ResNet18 to extract style latent as the condition vector. All experiments were conducted on the Linux server with four NVIDIA A100 GPUs. The noise $\xi_z$ is Gaussian noise with an amplitude of 0.001 by default. Detailed training hyperparameters for LoRA fine-tuning and COND P-DIFF framework are provided in Appendix B.

**Inference procedures.** In NLP tasks, we generate 20 LoRA parameters for each task using a conditional diffusion model through random noise and merge these generated parameters into the pre-trained model. We select the model that exhibits the best performance on the training dataset and report its performance on the validation dataset. In style-transfer tasks, we synthesize LoRA parameters of the corresponding styles by feeding the conditional diffusion model with images in various styles as conditions. We then merge parameters with PIXART-$\alpha$'s and utilize them to generate images using a set of prompts. Finally, we compute the FID score of the generated images.

**Baselines.** 1) **original**: The best validation performance among the originally trained models. 2) **model soup**: The validation performance of the model whose weight is the average of the training

dataset. Because Mitchell et al. Wortsman et al. (2022) shows averaging the weights of fine-tuned models with different hyperparameter configurations often improves accuracy and robustness. In style-transfer experiments, we introduce an additional baseline **no-lora**: we directly employ the predefined PIXART-$\alpha$ model to demonstrate the effectiveness of LoRA fine-tuning in style-transfer tasks.

## 4.2 EXPERIMENT RESULTS

**COND P-DIFF can generate high-performance parameters based on task conditions.** Table 1 presents comparison results of COND P-DIFF and baseline methods across language understanding GLUE benchmark for three models with different LoRA configurations. We observe that COND P-DIFF consistently yields comparable performance in most scenarios, demonstrating it learns conditional parameter distributions effectively and stably. Besides, we note that the baseline **average**'s performance in some cases surpasses the baseline, validating the potential of model averaging to enhance performance Wortsman et al. (2022).

Table 2 illustrates the results of COND P-DIFF and the baseline in the image style transfer task for different styles. We employ the FID Heusel et al. (2017) to quantitatively assess the quality of style-conditioned image generation. Lower FID represents better image generation quality. Based on our findings, COND P-DIFF efficiently synthesizes specific style-adapted LoRA parameters to generate high-quality images. Additional visual results are shown in Figure **??**. This demonstrates that COND P-DIFF can practically generate high-performance model parameters based on specific conditions.

Table 1: Results of COND P-DIFF on GLUE. We present results in the format of 'COND P-DIFF/ orginal / model soup'. COND P-DIFF obtains comparable or even better performance than baselines. 'Size' is the parameter size of LoRA. 'Rank' is the parameter $r$ in LoRA. Full' represents fully fine-tuning results.

| Model | Rank | Size | SST2 | RTE | MRPC | COLA | QNLI | STSB | Average |
|---|---|---|---|---|---|---|---|---|---|
| BERT | 1 | 73728 | **91.6** / 91.6 / 90.8 | 57.4 / **58.9** / 57.9 | **87.2** / 83.4 / 83.9 | 52.4 / **52.6** / 52.1 | **88.7** / 88.7 / 88.1 | **81.8** / 81.4 / 81.7 | **76.5** / 76.1 / 75.8 |
| | 2 | 147456 | **91.4** / 91.4 / 91.5 | 57.5 / 59.9 / **60.1** | **87.3** / 85.1 / 85.5 | **51.4** / 51.3 / 50.7 | **88.6** / 88.1 / 87.4 | **82.6** / 81.6 / 81.7 | **76.5** / 76.2 / 76.2 |
| | 4 | 294912 | 91.6 / 91.9 / **92.0** | 62.7 / **63.2** / 62.8 | 85.4 / 85.4 / **85.5** | **53.7** / 53.4 / 52.5 | **89.8** / 89.6 / 88.9 | 80.6 / **80.9** / 80.7 | 77.3 / **77.4** / 77.1 |
| | 16 | 1179648 | **92.1** / 91.6 / 91.5 | 64.2 / 64.3 / **64.5** | **87.4** / 87.0 / 86.8 | 56.9 / 57.0 / **57.5** | 89.8 / 90.1 / **90.2** | 83.8 / 83.3 / 82.3 | **79.0** / 78.9 / 78.8 |
| | Full | 109482240 | 93.5 | 66.4 | 88.9 | 52.1 | 90.5 | 85.8 | 79.5 |
| RoBERTa | 1 | 73728 | 93.3 / **93.7** / 94.1 | 65.6 / **68.6** / 68.0 | **86.9** / 84.7 / 85.0 | 49.8 / 50.2 / **50.5** | **92.4** / 92.0 / 91.4 | 87.3 / 87.5 / **86.9** | 79.2 / **79.4** / 79.3 |
| | 2 | 147456 | 93.5 / 93.7 / **93.8** | 63.2 / 68.2 / **68.3** | **87.7** / 85.0 / 84.6 | 50.3 / **50.7** / 50.6 | **92.8** / 92.5 / 92.2 | 86.8 / 87.3 / **87.6** | 79.0 / **79.6** / 79.5 |
| | 4 | 294912 | **93.8** / 93.5 / 93.1 | **69.8** / 69.7 / 69.5 | 87.9 / **88.3** / 87.9 | **54.1** / 54.0 / 54.1 | 92.0 / **92.4** / 92.9 | 88.3 / 88.2 / **88.6** | **81.0** / 81.0 / 81.0 |
| | Full | 124645632 | 94.8 | 78.7 | 90.2 | 63.6 | 92.8 | 91.2 | 85.2 |
| DeBERTa | 1 | 92160 | 94.4 / 94.4 / **94.7** | 61.4 / 61.0 / **61.5** | **84.0** / 84.0 / 83.2 | 56.8 / **57.0** / 56.1 | 92.4 / **92.8** / 92.1 | 87.4 / **87.8** / 87.0 | 79.4 / **79.5** / 79.1 |
| | 2 | 184320 | **94.9** / 94.8 / 94.0 | **62.2** / 62.1 / 62.0 | **86.2** / 85.8 / 86.2 | **58.6** / 58.3 / 57.4 | 92.1 / 92.0 / 92.1 | **85.2** / 85.2 / 84.5 | **79.9** / 79.4 / 79.4 |
| | 4 | 368640 | 94.6 / 94.5 / **94.7** | **63.2** / 62.8 / 61.9 | **87.1** / 86.9 / 86.2 | **60.3** / 60.3 / 59.9 | **93.4** / 93.5 / 93.1 | **88.7** / 88.7 / 88.7 | **81.2** / 81.1 / 80.7 |

Table 2: FID results of image-transfer tasks. Lower FID is better. Best results are **bolded**.

| Style | original | model soup | no-Lora | COND P-DIFF |
|---|---|---|---|---|
| Van Gogh | **27.92** | 28.08 | 102.95 | **28.03** |
| Edvard | **27.10** | 27.13 | 96.18 | **26.98** |
| Chalk | 36.22 | **36.00** | 171.82 | 36.18 |
| Charcoal | 40.80 | **40.19** | 132.76 | 40.60 |
| Average | **33.01** | 32.86 | 125.93 | 32.94 |

Table 3: Ablation results of training dataset size $N$. Larger $N$ can enhance performances.

| N | SST2 | STSB | MRPC |
|---|---|---|---|
| 1 | 90.23 | 80.71 | 82.71 |
| 100 | 91.63 | 80.91 | 83.52 |
| 200 | 91.63 | 81.81 | 87.24 |
| 500 | 91.63 | 81.80 | 87.25 |

## 4.3 ABLATION STUDY

In this section, we conduct multiple ablation studies to report the characteristics of COND P-DIFF. We focus on the performance of generated LoRA parameters(rank $r = 1$) of BERT on SST2, RTE, and MRPC datasets. The training setting is the same as experiments Table 1.

**Size of the training dataset** As described in Section 3.2, we collect $N$ different checkpoints in the last $N$ steps as a training dataset for task $q$ using LoRA. We explore the relationship between dataset size $N$ and performance in Table 3. We observe that the performance improves as the size of the

**Table 4: Ablation studies of COND P-DIFF. We ablate the normalization methods in the training process, the condition representation, and the location of employing COND P-DIFF. The default settings in COND P-DIFF are marked in** gray **. Bold entries are best results.**

a Comparison among no norm., batch norm., and task norm.. task norm. can improve performance.

| Norm. | SST2 | STSB | MRPC |
|---|---|---|---|
| no norm. | 55.67 | 49.07 | 47.01 |
| batch norm. | 90.60 | 80.90 | 82.50 |
| task norm. | **91.63** | **81.81** | **87.24** |

b Few-shot examples boost COND P-DIFF capability with task information description.

| Condition | SST2 | STSB | MRPC |
|---|---|---|---|
| one-hot | 90.05 | 77.12 | 80.34 |
| learnable vector | 90.10 | 80.03 | 81.81 |
| task info | 90.25 | 80.32 | 81.98 |
| task info+few-shot | **91.63** | **81.81** | **87.24** |

c COND P-DIFF is effective in certain blocks but can boost performance on whole LoRA parameters.

| LoRA layers | SST2 | STSB | MRPC |
|---|---|---|---|
| 0-1 | **91.63** | 81.43 | 83.45 |
| 0-4 | **91.63** | 81.45 | 83.61 |
| 0-8 | **91.63** | 81.80 | 85.61 |
| 0-11 | **91.63** | **81.81** | **87.24** |

training dataset increases. Specifically, a larger training dataset can provide a broader exploration space, thereby enabling COND P-DIFF to generate higher performance parameters. For instance, performance on the MRPC task improved by 4.53%.

**Normalization approach** As described in Section 3.2, we use *task normalization* method. Table 4a shows the impacts of different normalization strategies on performance, including *no norm.*, *batch norm.*, and *task norm.*. Specifically, *task norm.* refers to normalizing the parameters corresponding to each task individually. *batch norm.* represents batch normalization. The experimental setup in Table 4a is consistent with that of the experiment in Table 1. We find that *task norm.* consistently yields the best average performance. *no norm.* leads to the worst performance because the wide variance in weight distributions across different tasks and outliers hinders the convergence of the autoencoder. Besides, *batch norm.* performed inferior to *task norm.*, as it introduces spurious correlations among parameters across different tasks.

**Condition information** The representation of the condition critically affects generation results. We explore how to represent the task condition effectively to guide conditional parameter generation, as detailed in Table 4b. Our approach categorizes representations into four types: using one-shot vectors, using only the task description, using only two-shot examples, and using both the task description and two-shot examples. Table 4b shows that combining the task description with examples yields better outcomes, suggesting that in-context learning can provide more information to establish relationships with the weight parameters.

**Which part of parameters to synthesis** We generate LoRA parameters for all blocks by default in Table 1. To explore the effectiveness of COND P-DIFF on different blocks, we present the performance when generating LoRA parameters for only certain blocks. The experiments in Table 4c illustrate that the method is more effective when generating parameters for all blocks. We hypothesize that as the number of synthesized parameters increases, the model has a larger exploration space, thereby boosting performance. Conversely, performance is constrained by the exploration space and original parameters when focusing on only a subset of parameters.

## 4.4 ANALYSIS

In this section, we conduct a detailed analysis of COND P-DIFF. Specifically, we explore two critical questions: First, does COND P-DIFF merely replicate training data, or can it generate high-performance model parameters that are distinct from the originals? Second, does the generated parameter space of COND P-DIFF have generalizability?

**COND P-DIFF is not merely cloning model parameters.**

**Similarity vs. Performance** First, we calculate the $L_2$ distance between the generated and original parameters. Figure **??** illustrates the relationship between the similarity of the generated parameters and performance. We observe that COND P-DIFF attains various similarities and achieves better performance compared to original fine-tuned weights across various datasets.

**Parameter distribution** We employ t-SNE Van der Maaten & Hinton (2008) to analyze the distributions of generated parameters and original weights of fine-tuned models on datasets COLA, QNLI,

and STSB, as shown in Figures **??**. We observe that the distribution of generated parameters by COND P-DIFF significantly differs from the original parameters. The distribution of the original parameters can be viewed as following the trajectory of the optimization process. In contrast, COND P-DIFF generates novel high-performance parameters by learning the distribution of parameters. Besides, the high-performance parameters generated by COND P-DIFF are dispersed more broadly, underscoring the generative model's potential to identify novel high-performance parameters beyond traditional optimization pathways. Interestingly, the high-performance parameter distributions generated by COND P-DIFF for the three datasets are very similar, demonstrating the necessity of exploring the high-performance parameter space.

**Trajectories of COND P-DIFF process.** Figure **??** visualizes the generated parameters at different time steps during the inference stage using t-SNE Van der Maaten & Hinton (2008) to explore the generation process in the image style-transfer tasks. We display five trajectories initialized from five different random noises and present the model soup and the original model parameters. The parameters derived from the model soup are located near the original parameters. We observe that the generated parameters gradually approach the original parameters but ultimately maintain some distance from them, indicating that COND P-DIFF generates high-performance parameters that are distributed differently from the original parameters rather than directly replicating them. The variations in the trajectories also demonstrate the robustness of COND P-DIFF.

**Generalizability** We examine the generalization of the generated parameter space in the task of image style transfer. We select parameters, $\theta_{\text{style1}}$ and $\theta_{\text{style2}}$, generated by COND P-DIFF conditioned two distinct styles, style1 and style2. To interpolate between these styles, we compute a new set of parameters $\theta_{\text{interp}}$ as $\theta_{\text{interp}} = (1 - \lambda)\theta_{\text{style1}} + \lambda\theta_{\text{style2}}$, where $\lambda \in [0, 1]$ is the interpolation factor. Subsequently, we evaluate the effectiveness of $\theta_{\text{interp}}$ in style transfer. Figure **??** illustrates the visualization of images generated by interpolated parameters between Style-1 and Style-2. As $\lambda$ increases from left to right, the style gradually shifts towards Style-2. The continuous style change demonstrates the generalization of the generated parameter space. We also explore the generalization of the condition space in the Appendix C

# 5 RELATED WORK

**Diffusion models** Diffusion models Ho et al. (2020); Dhariwal & Nichol (2021); Peebles & Xie (2023) have recently emerged as a powerful class of generative models, enabling high-fidelity synthesis of complex data distributions. The research on the diffusion model can be generally classified into four categories. The first category aims to enhance image synthesis quality Rombach et al. (2022); Ramesh et al. (2022); Saharia et al. (2022) Second, researchers focus on accelerating the sampling process Song et al. (2022); Lu et al. (2022). Third, recent research has also focused on reevaluating diffusion models through the lens of continuous analysis like score-based generative modeling Feng et al. (2023). Fourth, the success of diffusion models has sparked their application in various domains, Kong et al. (2021); Luo & Hu (2021); Wolleb et al. (2022). In this work, we explore the conditional diffusion model in the parameter generation domain.

**Conditional generation** Conditional generation has gained significant attention in computer vision and natural language processing. Three prominent frameworks have emerged: conditional GANs Mirza & Osindero (2014); Isola et al. (2018); Zhu et al. (2020), conditional VAEs Sohn et al. (2015); Yan et al. (2016), and conditional diffusion models xwRombach et al. (2022); Ho et al. (2020), which incorporate conditions to guide the generation process, enabling the creation of visually coherent and semantically meaningful data samples. Conditional GANs incorporate condition information into GAN to generate images conditioned on specific attributes or labels. Conditional diffusion models take this further by generating visually coherent and semantically meaningful images from the textual description, demonstrating superior image synthesis quality compared to GANs. Building upon the success of conditional diffusion models, we propose to extend this approach to generating neural network parameters based on specific conditions.

**Parameter generation** The field of parameter generation has seen significant progress in recent years, with HyperNetworks ((Ha et al., 2016) and generative models of neural network checkpoints Peebles et al. (2022) emerging as promising approaches. Ha et al. (2016) introduced HyperNetworks, which uses a hypernetwork to learn the parameters for another neural network. Finn et al. (2017) proposes Model-Agnostic Meta-Learning, which learns an initialization for efficient fine-tuning.

Peebles et al. (2022) introduce the model G.pt to predict the distribution over parameter updates given an initial input parameter vector and a prompted loss or error. Schürholt et al. (2022) trained autoencoder on a model zoo to learn a hyper-representation for generative use to sample new model weights Knyazev et al. (2021) use a GNN-based model to sample network parameters. Erkoç et al. (2023) directly leverages MLP weights and generates neural implicit fields encoded by synthesized MLP weights. Wang et al. (2024) uses a diffusion model to generate high-performing neural network parameters across various architectures and datasets. Different from the previous works, we focus on conditional parameter generation to generate high-performing weights based on specific task conditions practically.

## 6 CONCLUSION

In this work, we proposed an approach COND P-DIFF for high-performance controllable parameter generation, specially for LoRA parameters. We utilize an autoencoder and a conditional latent diffusion model to capture the distribution of high-performing parameters and perform conditional generation, synthesizing a new set of parameters tailored to specific conditions. We show that our method can efficiently synthesize novel and high-quality model parameters. The parameter distribution generated by COND P-DIFF exhibits differences compared to the distribution obtained through conventional optimization methods, indicating a certain level of generalization capability.

### 6.1 LIMITATION AND FUTURE WORK

Nonetheless, it is essential to recognize that diffusion in parameter generation is still largely unexplored despite the significant advances in the realm of image and video synthesis. In this work, we present a preliminary methodology for conditional parameter diffusion. However, several challenges remain unresolved, including reducing memory demands for large model architectures, enhancing the generalizability of generation techniques, and improving the representation of dataset conditions. Furthermore, integrating knowledge graphs with conditional diffusion offers promising directions for controlling conditional generation.

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
