# OpenReview forum: "Conditional LoRA Parameter Generation"
_ICLR.cc/2025/Conference — ICLR 2025 Conference Withdrawn Submission_

### Official Review · Reviewer_hSao · 2024-10-29

**Soundness:** 3
**Presentation:** 3
**Contribution:** 3
**Rating:** 5
**Confidence:** 4

**Summary:**

The paper proposes a method to generate LoRA parameters using a conditional latent-diffusion process. It studies the setup for language understanding and image stylization tasks. COND P-DIFF shows performance that is either comparable to the performance of the LoRA params obtained through conventional optimization process or sometimes outperforms them.

**Strengths:**

* The paper goes beyond shallow architectures, unlike existing works in the area, and presents results with network architectures that are widely used. Moreover, it uses pre-trained networks and studies parameter generation for LoRAs, which makes it practically compelling.
* The overall methodology is simple, and the components involved have been studied well in the paper and in the literature.
* Studies two different modalities (language and images) that make it appealing.

**Weaknesses:**

* Even though the paper studies two modalities, the number of tasks studied seems limited as far as practicality is concerned, which is a central theme of this work. For the image domain, it studies only image stylization but misses out on other potentially more practical tasks that may require more samples as conditions (image classification, for example).
* Comparisons between LoRA fine-tuning time and LoRA parameter generation time (including the COND P-DIFF training time) seem to be missing. So, perhaps a time vs. performance trade-off would help make the argument of caring more about approaches like COND P-DIFF stronger.
* Authors flatten the low-rank weight matrices to construct their datasets. For very large parameter spaces, this can quickly become memory-expensive.

**Questions:**

* The abstract could include how the base network is employed during the entire process, as the LoRA params are supposed to be tied to the base network.

* L021 - L023: It could be specified more directly that the differences between the params (COND P-DIFF vs. regular) are tied to generalizability. Or maybe the sentence could be rewritten a bit to denote the kind of differences.

* L054 - L055: Could it be that the scalability approach wasn't studied in those respective works because it was relatively new back then? Or is it that the methods themselves aren't particularly suitable for scaling?

* L058: "Simple architectures" such as?

* L060 - L061: The sentence could be expanded a little bit to specify "... do not support task-conditioned parameter generation"? Also, there seems to be a minor formatting issue in the citation (Wang et al. (2024) should go into brackets?) This applies elsewhere, too, throughout the text.

* L088 - L092: Figure 2 could be cited here, stating that a pictorial representation of the methodology is available.

* L092: The authors mention that the diffusion model reconstructs the latent representations. But shouldn't the representations then go to the decoder of the previously trained Autoencoder to obtain the final parameter-space representations? I see that Figure 2 already specifies it and the associated text. So, perhaps the text in this section could include it, too?

* L096 - L097: One/two examples of tasks/datasets could be provided beside the point where the authors claim that COND P-DIFF achieves superior performance.

* L100: I think the third point is a combination of the earlier two points and could be avoided. Instead, the authors could consider including examples of tasks/datasets they considered in the work and the methods used for comparing the parameter representations.

* Equation 5 could include the approximation typically followed in the diffusion world, i.e., the "epsilon-prediction" objective [1].

* L162: Are these base model parameters or existing LoRA parameters? The Figure 2 caption states that it's the LoRA parameters, but I think the main text also should.

* In the second block of Figure 2, $z_T$ could directly go to the "Denoising Net" block? Otherwise, the diffusion process seems a little disconnected from the denoising net.

* I understand that the dataset preparation methodology allows for more controlled experiments, but I think a good value addition of COND P-DIFF could be leveraging existing task-specific LoRA parameters made available by the community (available through the Hugging Face Hub). I can also imagine that some LoRA parameters might be corrupted. So, to mitigate the risk, only popular ones could be considered. I encourage the authors to explore this avenue.

* L199 - L200: If possible, the Z-score normalization could be expanded in the context COND P-DIFF.

* L196: Could the dimensions of each $\theta_n$ be specified too?

* For the autoencoder:
  * How much does the inclusion of Gaussian noise help?
  * Did the authors try out a VAE in this setup, as that is what's more commonly used in the latent diffusion literature?

* L222: There is no decoder involved in CLIP.

* L223: How much does the choice of the number of examples in the in-context learning matter for the NLP tasks?

* The CLIP image encoder could be leveraged to extract the style representations or general image-level representations.

* I request the authors to consider more practical tasks for computer vision, such as image classification, which allows
us to study how many images we need to achieve a reasonable performance with COND P-DIFF. For many image-level tasks (such as classification), we need more training images than we need for image stylization.


* L262: Why generate 20 LoRA parameters? Is this number sufficiently ablated? Does it incur significant latency?

* L264: The text says that the parameters are merged into the base model. So, I think what the process follows is that for each 20 generated LoRA parameters, it's first merged into the base model and, the final model is then evaluated.

* L289: Figure citation is wrong. This applies elsewhere too.

* L361: "all blocks" should be specified. Did the authors target only the KV layers in the attention modules (as done in the original LoRA paper) or something else?

* For comparing the LoRA parameter representations, a framework like CKA could be used as it is more robust and resilient compared to L2.

* Do the LoRA parameters obtained through COND P-DIFF show similar behavior during inference when their scales vary (the LoRA scale helps to control how much the LoRA params should contribute)?

**References**

[1] Denoising Diffusion Probabilistic Models; Ho et al.; 2020.

---

### Official Review · Reviewer_nBav · 2024-11-01

**Soundness:** 2
**Presentation:** 2
**Contribution:** 2
**Rating:** 3
**Confidence:** 4

**Summary:**

To generate task-specific parameters for pretrained models, this paper proposes using a latent diffusion model to synthesize LoRA weights. Specifically, the authors collect LoRA weights for adapting pretrained models to various tasks and introduce a task-norm process to standardize the data. Following stable diffusion training strategies, they then train the model to synthesize task-specific parameters based on text or image inputs. Experiments demonstrate that the method can generates task-specific data and achieves good performance.

**Strengths:**

1. The authors propose a new approach using a latent diffusion model to synthesize LoRA weights for pretrained models.

2. The authors gather a dataset of LoRA weights tailored to specific pretrained models, using it to train the latent diffusion model.

3. Experiments on both language and image tasks show that this method can synthesizes LoRA weights for pretrained models, achieving good performance.

**Weaknesses:**

1. The dataset size is small, which risks the model memorizing the data. While the authors show that the synthesized parameters differ from the training data, the diffusion process may only add Gaussian noise to the parameters training dataset. Additionally, the authors' use of averaging synthesized weights for specific tasks may eliminate Gaussian noise effects in diffusion process. They should present performance results without averaging.

2. The claim that the proposed method generalizes well is overstated. The authors demonstrate generalizability using images used for collecting the parameter training dataset, which is unconvincing. Testing with new images not used for collecting the training set would strengthen the claim.

3. Related work is covered in both the main paper and the supplementary material, which could be combined. Additionally, some figure references are missing in the main paper.

4. The style transfer results are weak.

**Questions:**

Please see the weaknesses.

---

### Official Review · Reviewer_FiZz · 2024-11-02

**Soundness:** 2
**Presentation:** 1
**Contribution:** 2
**Rating:** 3
**Confidence:** 4

**Summary:**

The paper introduces "COND P-DIFF," a novel framework for generating task-specific model parameters using parameter-space conditional latent diffusion models, particularly the LoRA (Low-Rank Adaptation) parameters. The authors first train an autoencoder to encode LoRA parameters into a latent space. The diffusion model is conditioned based on a domain specific encoder. For instance, CLIP Text Decoder with description of task and examples in case of NLP tasks and ResNet features of an image in case of visual domain. The authors show results on a wide range of NLP tasks and image stylization task where COND P-DIFF outperforms the original model.

**Strengths:**

The idea of generating LoRA parameters with a diffusion model is quite interesting. Overall, the paper is well written and clearly explained. The experiments and results section is comprehensive with results on various tasks.

**Weaknesses:**

1. One major concern is the absence of the corresponding figures in Section 4.4 (Analysis). It is not possible to make any inferences about the experiments without the referenced figures. And I think Section 4.4 is important to understand whether the diffusion model is learning anything new compared to the original LoRA weights.
2. One important baseline that would be useful in this setup is to use randomly select one of the training checkpoint and analyse it for a new test image (in case of style transfer). Similar thing can be done in the case of NLP tasks, where LoRA checkpoint for one task can be used for the other task. Ideally, one should observe a drop in performance when this happens.
3. One of the biggest motivations for learning a generative model in the parameter space was ensembles (as discussed in [1]). In this case of LoRA parameter, the authors generate 20 LoRA parameters from the diffusion model and then select the best one. The authors should explore other ways of using the parameters potentially.
4. The writing of this paper needs to improve massively.

- L269-270 sentence is unclear. I guess the authors mean to take the average of weights of various finetuned models (LoRA).
- In Section 4.4: Analysis, none of the referenced figures are present in the paper.
- L420 has `xw`  for some reason.

[1] Wang, Kai, et al. "Neural network diffusion." arXiv preprint arXiv:2402.13144 (2024).

**Questions:**

1. The authors should discuss and elaborate on the cost of training the diffusion model and the autoencoder. How long does it take to train both these models?
2. In L249-250, it is mentioned that “we collect 150 training samples for models, including BERT, Roberta, GPT-2 by default”. I assume that this is 150 models for each task. Since a single conditional diffusion model is trained across tasks, there are a total of 900 checkpoints in the dataset (150 * 6 tasks) for training the diffusion model. If this is true, have the authors evaluated the performance on a OOD task (a task that was not present in the training dataset) ?

---

### Official Review · Reviewer_hiQL · 2024-11-03

**Soundness:** 2
**Presentation:** 2
**Contribution:** 3
**Rating:** 5
**Confidence:** 3

**Summary:**

This paper proposes COND P-DIFF, a new method for generating LoRA parameters for model fine-tuning. This approach employs a conditional latent diffusion model to generate LoRA parameters based on specific task conditions, using an autoencoder for efficient latent representation. COND P-DIFF is evaluated in both computer vision and natural language processing tasks, demonstrating that it can produce task-adapted parameters that achieve performance comparable to traditional optimization methods.

**Strengths:**

1.	Originality: The idea of using conditional diffusion models to synthesize LoRA parameters for fine-tuning is novel, providing an innovative method for model adaptation. To the best of my knowledge, this is the first paper that bridges parameter-efficient tuning methods and conditional diffusion parameter generation.
2.	Quality: The method is well-defined, with experiments that confirm the approach’s effectiveness. Comparisons with traditional fine-tuning and model-averaging techniques are included to highlight the method’s advantages. Additionally, an ablation study explores the impact of training dataset size, normalization techniques, and LoRA layers on performance.
3.	Clarity: Overall, the paper is well-structured, with clear definitions of methods and explanations of experiments. However, certain sections could benefit from clarification, such as the autoencoder design, which is not fully elaborated.
4.	Significance: The approach offers a meaningful advance in parameter generation for LoRA, which is highly valuable for real-world applications requiring adaptive, low-resource methods.

**Weaknesses:**

1.	Some figures (e.g., result comparisons and t-SNE analyses of parameter distributions) are missing, limiting the reader’s ability to evaluate the quality and diversity of generated parameters visually. The appendix is also missing.
2.	While the paper mentions using several datasets (e.g., GLUE, SemArt, WikiArt), specific citations and descriptions are absent. This omission makes it challenging for readers to understand the dataset characteristics and reproducibility. Additionally, results on these datasets are only summarized without detailed visualizations, which would improve interpretability.
3.	Although an autoencoder is used to reduce the dimensionality of LoRA parameters, training it adds complexity to the pipeline compared to traditional optimization. More information on the efficiency gains it provides or alternative dimensionality reduction techniques would strengthen this approach’s feasibility.
4.	Some details in the paper need to be revised. For example, in Table 1 the quotation marks are not in the correct format. In Table 2 the method with lowest average FID is not marked correctly.

**Questions:**

1.	Can the authors clarify how different dataset characteristics affect the generated parameters? Including more diverse datasets and citation information would provide context and allow readers to appreciate the model’s flexibility across domains.
2.	Missing visual comparisons (such as t-SNE plots) of generated vs. real parameters hinder the interpretation of the generalization claims. Adding these would substantiate the discussion on the generative capacity of COND P-DIFF.
3.	Could the authors discuss how the condition would affect the performance? Compared with traditional training, generating LoRA weights can use only two examples as the condition to reach a similar or even stronger performance than training, which is not that easy to understand.
4.	Could the authors provide more details on the training pipeline of the model and the role of the autoencoder in terms of efficiency and performance? Whether the training of the model need multiple steps or end-to-end?

---

### Official Review · Reviewer_8Va7 · 2024-11-03

**Soundness:** 2
**Presentation:** 1
**Contribution:** 1
**Rating:** 1
**Confidence:** 5

**Summary:**

The authors propose a LoRA parameter diffusion models conditioned on specific image and NLP tasks.

**Strengths:**

1. The presented qualitative results are on par or slightly better than baselines.

**Weaknesses:**

1. The paper seems very rushed and incomplete. Figures are missing even though there are references in the text. As a consequence, no visual results are provided, nor other analysis claimed in the paper.
2. The novelty is very limited. The authors build on top of P-Diff by adding only a task conditioning mechanism.
3. There are numerous editorial errors, e.g. lack of spaces (lines 061, 222, 292), wrong formatting (eq. 6b, 7), grammatical errors (270-271), Fig vs Figure, etc.

**Questions:**

1. Why is conditional parameter generation important? The authors state this as a fact but the explanation (even a short one) is missing.
2. Could you explain what is the conditioning for both NLP and image tasks?
3. The related work section should be placed after introduction.

---

### Note · Authors · 2024-11-14

I have read and agree with the venue's withdrawal policy on behalf of myself and my co-authors.